# LOGIT ATTENUATING WEIGHT NORMALIZATION

## ABSTRACT

Over-parameterized deep networks trained using gradient-based optimizers is a popular way of solving classification and ranking problems. Without appropriately tuned regularization, such networks have the tendency to make output scores (logits) and network weights large, causing training loss to become too small and the network to lose its adaptivity (ability to move around and escape regions of poor generalization) in the weight space. Adaptive optimizers like Adam, being aggressive at optimizing the train loss, are particularly affected by this. It is well known that, even with weight decay (WD) and normal hyper-parameter tuning, adaptive optimizers lag behind SGD a lot in terms of generalization performance, mainly in the image classification domain.

An alternative to WD for improving a network's adaptivity is to directly control the magnitude of the weights and hence the logits. We propose a method called Logit Attenuating Weight Normalization (LAWN), that can be stacked onto any gradient-based optimizer. LAWN initially starts off training in a free (unregularized) mode and, after some initial epochs, it constrains the weight norms of layers, thereby controlling the logits and improving adaptivity. This is a new regularization approach that does not use WD anywhere; instead, the number of initial free epochs becomes the new hyper-parameter. The resulting LAWN variant of adaptive optimizers gives a solid lift to generalization performance, making their performance equal or even exceed SGD's performance on benchmark image classification and recommender datasets. Another important feature is that LAWN also greatly improves the adaptive optimizers when used with large batch sizes.

## 1 INTRODUCTION

The advent of large scale deep models with tens of millions to billions of parameters has resulted in three trends in the community: (1) State-of-the-art performance via over-parameterized networks for problems like image classification, language modeling, machine translation, text classification and recommender systems. (2) Development of optimizers like Stochastic Gradient Descent (SGD) with heavy-ball momentum (Qian, 1999), Adam (Kingma & Ba, 2017), AdamW (Loshchilov & Hutter, 2019), LAMB (You et al., 2020) and their extensions with weight decay/$\ell_2$ regularization to improve generalization performance. (3) Increased emphasis on theory to understand the optimizer landscape, especially how to tune different hyperparameters well to escape poor minima for both large and small batch sizes. In this work, inspired by all three trends, we propose a new training method, explain why it works, and show vastly improved performance over the weight decay method when applied with adaptive optimizers across a wide range of batch sizes for classification and ranking tasks.

Complex deep networks can easily learn to classify a large fraction of examples correctly as training progresses. These networks have two characteristics: (a) they have one or more contiguous *homogeneous*[1] layers at the end forming the end homogeneous sub-network, and, (b) they are trained with exponential-type loss functions like logistic loss and cross entropy that asymptotically attain their least value of zero when the network score goes to infinity. Let us collectively refer to this network score as *logit*. After the network has learned to correctly classify a large fraction of training examples, the weights of the end homogeneous layers and the logits grow to make the training loss (and hence its gradient) very small. This, seen in optimizers when used with no (or mild) weight decay/$\ell_2$ regularization, leads to *loss flattening* (loss and gradient taking very small values) (see Appendix A). This further leads to *loss of adaptivity* of the network (Szegedy et al., 2015), causing training to stall in regions of sub-optimal generalization (see §2).

---

[1]A layer is *homogeneous* if the activation function of each unit of the layer satisfies $\phi(\alpha x) = \alpha\phi(x)$. Linear and ReLU are examples of such activation functions.

Figure 1 shows the generalization performance (Test HR@10) of Adam without weight decay (green dotted line) applied to a fully homogeneous network on a classification task. After about 115 epochs, *Margin p50* (median margin over the training examples; blue dotted line) becomes large and the training gets stuck in the basin of a sub-optimally generalizing minimum. The minimum is also overfitting due to the tussle between good and noisy examples.

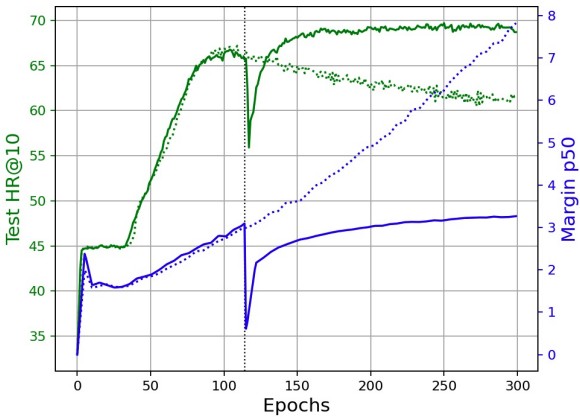

**Figure 1:** Adam with (continuous lines) and without (dotted lines) logit attenuation on MovieLens classification. *Adam:* after 115 epochs (vertical dotted black line), loss flattening sets in, Test HR@10 is sub-optimal, with overfitting. *Adam with logits attenuation:* The introduction of $\alpha$ factor after the 115th epoch reduces *Margin p50* but keeps Test HR@10 the same. Though Test HR@10 then drops initially, with further constrained weight training Adam with logit attenuation eventually achieves a higher Test HR@10 than vanilla Adam.

| Optimizer | Batch Size | |
| --- | --- | --- |
| | 256 | 16k |
| SGD | 76.00 (0.04) | 74.48 (0.06) |
| Adam | 71.16 (0.05) | 70.60 (0.02) |
| Adam-L | **76.18** (0.03) | **76.07** (0.08) |
| LAMB+ | 74.30 (0.01) | 73.43 (0.03) |
| LAMB-L | **76.48** (0.05) | **75.93** (0.02) |

**Table 1: ImageNet validation accuracy.** Comparison of SGD, Adam and LAMB (all with WD) and LAWN variants (*-L) of Adam and LAMB+ [a] on the ImageNet validation set. LAWN enables Adam to work on image classification tasks with very little drop in performance at large batch sizes. Current optimizers have a much steeper drop-off in performance as batch size increases. Standard error is mentioned in parentheses.

---

[a]LAMB+ is a modification of the LAMB algorithm. More details can be found in Appendix B.3.3.

Adaptive gradient algorithms like ADAM are particularly severely affected by loss flattening. These algorithms are based on tracking the local loss behavior at each individual weight level and hence they achieve fast convergence. However, in domains such as image classification, their generalization performance quickly plateaus to a value that is much worse than the performance finally achieved by SGD. This gap is usually attributed to the inability of adaptive gradient algorithms to escape the basin of a sharp minimum that has large curvature and possibly poor generalization. While weight decay helps improve adaptive optimizers, it is still not sufficient to close the gap with SGD. The training method proposed in this paper closes the gap.

To motivate the method, let us return to the experiment of Figure 1. Consider that training has reached a point where logits are starting to become large. Attenuation of the logit values can be done using the following two ideas: (a) multiply the logits by a factor, $0 < \alpha < 1$ and use this factor for the rest of the training; (b) constrain the norms of layer weights for the rest of the training. Specifically, in the experiment of Figure 1, after epoch 115 we shrink the logits to one-fifth, and then continue training while keeping the weight norms of each layer fixed. This leads to superior generalization with no overfitting (see the continuous lines in the figure). It turns out that idea (a) alone is not sufficient since further training will increase the logits again to reduce the training loss; so, idea (b) is very important. Also, in order to control the logit magnitudes, instead of idea (a) (waiting till 115 epochs and then reducing the logits by a factor $\alpha = 0.2$) we could simply start using idea (b) without applying an $\alpha$ factor, at an earlier point, say, after just 5 epochs.

This lays the base for our method, Logit Attenuating Weight Normalization (LAWN). It can be used with any gradient-based base optimizer, though our aim is mainly to improve adaptive optimizers. LAWN begins with normal, unconstrained training in the initial phase and then constrains the weight norms of the layers for the rest of the training. Weight decay is not used anywhere. The training on the constrained norm surfaces is done by employing projected gradients instead of regular gradients.

The LAWN variants of Adam and LAMB achieve impressive performance across multiple network architectures and tasks. At large batch sizes, most optimizers get caught in sub-optimal regions due to lowered stochasticity which is worsened by increased logits. LAWN's attenuation of the logits

helps avoid this worsening. Due to this, the LAWN variants of Adam and LAMB work significantly better than their base versions at large batch sizes. Table 1, which compares the generalization performance of SGD, Adam and LAMB (all with weight decay) against the LAWN variants of Adam and LAMB on the popular Imagenet dataset, powerfully showcases the two observations on LAWN mentioned above. In §4 we show that these LAWN variants give much better generalization than weight decay on several architectures for image classification (CIFAR, ImageNet) and recommender systems (MovieLens, Pinterest). We end this section with a discussion of related work.

**Related work**: The issue of loss flattening and the resulting loss of adaptivity is highlighted in (Szegedy et al., 2015). Several well-known techniques are used to mitigate this issue. $\ell_2$ regularization is often used in conjunction with optimizers like SGD and adaptive optimizers (Kingma & Ba, 2017; Zeiler, 2012; Duchi et al., 2011), among others. Recently, decoupled weight decay (Loshchilov & Hutter, 2019; Hanson & Pratt, 1988) has become popular to reign-in network weights and prevent overconfidence on training samples. Other techniques are: (a) label smoothing regularization (Szegedy et al., 2015), which makes the model less confident about predictions by changing the ground-truth label distribution; and (b) flooding (Ishida et al., 2020) which tries to keep the aggregate training loss to be around a specified small value. From a theory perspective, weight norm bounding has been shown to be useful for improving generalization (Neyshabur et al., 2015; Bartlett et al., 2017). Salimans and Kingma (Salimans & Kingma, 2016) use weight normalization as a transformation; but, by also keeping the scale component, they end up allowing logits to grow large. Hoffer et al (Hoffer et al., 2018) discuss keeping the norms of the parameters fixed, but the method would require a LAWN-like approach to work effectively.

It has been observed that small batch sizes yield better generalization performance (Keskar et al., 2016) due to the noise of mini-batch gradients and the large learning rate. The noise diminishes with increasing batch sizes. Large batch sizes are useful for speeding up the training process by leveraging parallel GPUs. Poor generalization for large batch sizes is attributed to them stalling around "sharp" minimizers (Keskar et al., 2016). Goyal et al (Goyal et al., 2018) scale ImageNet training to batch sizes of 8k without loss in generalization performance, by carefully tuning parameters like learning rate and batch normalization. Other recent efforts to train large-batch models include (Hoffer et al., 2017; You et al., 2017; 2019; Shallue et al., 2018; You et al., 2020). It is important to note that for a majority of the cited works, large batch gains do not necessarily hold across tasks or datasets.

## 2 THE NEED FOR LAWN

In this section we motivate LAWN by describing the problem settings in which normal training struggles and in which LAWN could improve generalization performance. We define these problem settings in §2.1, then describe what we mean by *loss of adaptivity* in §2.2. We review the strengths and weaknesses of existing methods for avoiding loss of adaptivity in §2.3. With this context, we elaborate on the LAWN method in §3.

**Notations.** For the rest of the paper we will use multi-class setting as the running example. The following notations will be used: $n$ is the number of weight variables; $m$ is the number of training examples; $i$ is the index used to denote the index of a training example; $y_i$ is the target class of the $i$-th training example; $k$ is the index used to denote a class; $nc$ is the number of classes; $w$ is the weight vector of the deep net; $p_k(w)$ is the class $k$ probability assigned by the deep net with weight vector $w$. For an optimizer, $\eta$ denotes the learning rate and $B$ is the batch size.

### 2.1 WHEN DOES LAWN WORK?

Let us describe the problem settings in which we expect generalization improvement using LAWN.

**Problem type.** We are mainly interested in classification and ranking problems which form a score for the target. In binary classification this score is the logit score of the target class; in a multiclass problem, this score is the difference between the target class score and the maximum score over the remaining classes. For ease of presentation we will simply refer to such scores as *logit*. We consider a loss function that attains its least value (usually zero) asymptotically as logit goes to infinity. The logistic loss for binary classification and the cross entropy loss based on softmax for multi-class classification are important cases. Problems like regression that use a loss function such as the squared loss (which attains its minimum at a finite value) may not have much benefit using LAWN. In a multi-task setting, when the total loss is an additive mix of several individual task losses, LAWN can be useful even if just one of them is suited for LAWN.

**Network Complexity.** This refers to the complexity of the network in relation to the number of training examples. Most deep networks used in applications are *over-parametrized*. Roughly, we will take *over-parametrized* to mean that the network is so powerful that training easily locates a $w$ that classifies most examples correctly, i.e., the target class has the highest score among all classes. Deep nets usually have one or more fully connected homogeneous layers (usually with ReLU units) at the end. If the deep net is powerful enough to correctly classify most examples correctly, then by making the weights large it is possible to push the loss of most training examples (and thereby the train loss) to very small values.

**Generalization Metrics.** Our focus is on improving metrics that are based on score ordering as opposed to the actual values of scores. More precisely, we are interested in test set metrics such as classification error, AUC, NDCG etc. as opposed to logistic loss, cross entropy loss, probability calibration error etc. Deep networks are known to be poor with respect to the latter metrics (Guo et al., 2017) but which can be improved in the post-training stage; the adaptation of LAWN to improving such metrics will be taken up in a follow-up work.[2]

### 2.2 ISSUE OF LOSS OF ADAPTIVITY

Consider the problem setting defined in §2.1 and the minimization of the normally used unregularized training loss, $L(w) = -\frac{1}{m}\sum_{i=1}^{m} \log p_{y_i}(w)$. With the network being sufficiently powerful, training causes the losses of a large fraction of the training examples to become very small. During training, this happens due to weights becoming large, the logit becoming large, and $p_{y_i} \to 1$ for those examples. Due to these, each of the following become very small: the gradient of most example-wise loss terms; $\Sigma$, the covariance of the noise associated with minibatch gradient; and, $H$, the Hessian of the training loss. We will refer to this collective happening as *loss flattening*.

It has been established via theoretical and empirical arguments that the powerful generalization ability of a deep net comes from its ability to escape regions of attraction of "sharper" minima[3] with sub-optimal generalization performance and go to better solutions. Appendix C gives an idea of the escape mechanism for the SGD method using a simplistic analysis given by Wu et al (Wu et al.), which we use just for guidance and motivation. It is worth recalling from there, the following rough guiding condition for SGD to escape from a poor solution: $\lambda_{\max}\{(I - \eta H)^2 + \frac{\eta^2(m-B)}{B(m-1)}\Sigma\} > 1$, where $\lambda_{\max}(A)$ denotes the largest eigenvalue of $A$. If training is at a sub-optimal generalization point, loss flattening occurs (which means that $H$ and $\Sigma$ become small), then the escape condition is difficult to satisfy and hence it becomes difficult for the network to escape from this solution and then train further to go to a better solution. A carefully increased learning rate schedule to cause the escape followed by the use of normal learning rates to go to a better solution can make this happen, but no such automatic sophisticated learning rate adjustment mechanism has been devised yet. We will refer to this inability of the network to escape out of a sub-optimal solution due to loss flattening as *loss of adaptivity* (also see §7 in (Szegedy et al., 2015)).

### 2.3 CURRENT METHODS FOR DEALING WITH LOSS OF ADAPTIVITY

Several methods have been suggested in the literature to handle the issue of loss of adaptivity. We briefly describe three key ones: label smoothing regularization (LSR) (Szegedy et al., 2015), flooding (Ishida et al., 2020), and $\ell_2$ regularization/weight decay (Loshchilov & Hutter, 2019). LSR modifies $L(w)$ via making the target class less confident by fixing its probability as $(1 - \epsilon_{LSR})$ and reassigning $\epsilon_{LSR}$ to the remaining classes. This makes the loss attain its minimum at finite logit values. Flooding modifies the loss as $L_{Flooding}(w) = |L - \epsilon_{Flooding}|$ so that the training process is forced to move around the hypersurface defined by $L - \epsilon_{Flooding} = 0$. $\ell_2$ regularization is a traditional method that modifies the loss as $L_{\ell_2} = L(w) + \frac{\lambda}{2}\|w\|^2$. (Decoupled) Weight decay, as used in the recent deep net literature decouples the term $\lambda w$ from the gradient and instead includes the additive term, $-\lambda w$ at the weight update step. In the next section (§3) we will return to discuss these methods in relation to LAWN.

## 3 THE LAWN METHOD

As we mentioned in §2, we consider deep nets that have one or more homogeneous layers (usually with ReLU units) at the end (top); let $fhsn$ refer to this *final homogeneous sub net*. Consider a weight

---

[2]However, as we show in §4.6, LAWN does much better calibration than the weight decay method.

[3]Originating from the work of Keskar et al (Keskar et al., 2016), it has been observed in the literature that minima at which the classification function has a sharper behavior has poorer generalization.

vector $\bar{w}$ and corresponding *fhsn* weights $\bar{w}_{fhsn}$. When we extend $\bar{w}_{fhsn}$ along a radial direction, i.e., $w_{fhsn} = \alpha \bar{w}_{fhsn}, \alpha \in R_+$, while keeping the weights of the network outside *fhsn* unchanged, the classification error on a set of examples (e.g., the training error computed on a training set or the generalization error computed on a test set) remains unaffected whereas classification loss varies a lot with $\alpha$. In particular, if $\bar{w}$ classifies a set of examples strictly correctly, then, as $\alpha$ goes from 0 to $\infty$, the average loss on these examples goes from $\log(nc)$ to zero asymptotically. When training without any/sufficient weight decay, weights do get large and training loss does become very small. In the previous section (§2) we saw how, when training loss becomes very small, it causes loss flattening, which leads to a loss of adaptivity of the network. Therefore, it makes sense to suitably contain the size of the weights. Theory (Neyshabur et al., 2015; Bartlett et al., 2017) suggests that, for improving generalization, it is a good idea to bound the weights of the entire network. Given that most layers are usually homogeneous[4] (e.g., units with ReLU activations), it is appropriate to use layer-wise weight normalization. The essential spirit of LAWN is along this idea.

Consider constrained training via layer-wise weight normalization: $\|w^\ell\| = c^\ell \ \forall \ell$ where $w^\ell$ is the weight vector accociated with layer $\ell$, and the $c^\ell$ are some constants. It is useful to understand how the loss contours behave as the $c^\ell$ go from small to big values. When the $c^\ell$ are large, we know that loss flattening will happen. When the $c^\ell$ are small, the distinction between the loss values of well classified examples and poorly classified examples diminishes and so, optimizers will find it harder to traverse the contours and go to the right place of best generalization. To choose the right $c^\ell$ values, in LAWN we take a simple and natural approach. We initialize the network with weights having small magnitude using a standard weight initialization method and start a given optimizer in its free (unconstrained, without any weight decay) form. At a suitable point in that training process, with weights at some $\bar{w}$, we switch to constrained training defined by setting

$$\|w^\ell\| = c^\ell \text{ where } c^\ell = \|\bar{w}^\ell\| \ \ \forall \ell \tag{1}$$

The $c^\ell$ are fixed for the rest of the LAWN training. Constrained training corresponds to solving the optimization problem,

$$\min L(w) \text{ s.t. } \|w^\ell\| = c^\ell \ \forall \ell \tag{2}$$

using a modified version of any given gradient-based optimizer.

In LAWN, we switch from free training to constrained training after some $E_{free}$ epochs, and tune $E_{free}$ as a hyperparameter using a coarse grid. Thus, when compared to regularization using weight decay, LAWN (a) does not use weight decay anywhere, and (b) it uses $E_{free}$ as a hyperparameter instead.

In the future, we plan to try the following automatic method for choosing $E_{free}$. Track the logits of training examples (done simply and efficiently by tracking them on the minibatches used) and switch to constrained training when their median value starts reaching high values indicating the onset of loss flattening.

---

**Algorithm 1** Adam-LAWN Constrained Phase

1: **for** t in 1...T **do**
2:      Draw batch $S_t$ from the training set
3:      $g_t = \text{ComputeGrad}(w_{t-1}, S_t)$
4:      Project each $g_t^\ell$ to $\{d^\ell : (w^\ell)^T d^\ell = 0\}$ to get $g_{pt}^\ell \ \forall \ell$
5:      $m_t = \beta_1 m_{t-1} + (1 - \beta_1) g_{pt}$
6:      $v_t = \beta_2 v_{t-1} + (1 - \beta_2) g_{pt}^2$
7:      $\hat{m}_t = \frac{m_t}{1 - \beta_1^t}, \hat{v}_t = \frac{v_t}{1 - \beta_2^t}$
8:      Compute $r_t = \frac{\hat{m}_t}{\sqrt{\hat{v}_t} + \epsilon}$
9:      Project each $r_t^\ell$ to $\{d^\ell : (w^\ell)^T d^\ell = 0\}$ to get $\hat{r}_t^\ell \ \forall \ell$
10:     $w_t = w_{t-1} - \eta_t \hat{r}_t$
11:     Rescale $w_t$ to satisfy constraints on $\|w^\ell\| \ \forall \ell$
12: **end for**

Hoffer et al (Hoffer et al., 2018) give a bounded weight normalization method which also constrains the weight norms and uses simple heuristics for setting the $c^\ell$. However, it does not use the idea of combining free and constrained training, and it does not set up and demonstrate the use of constrained training as a powerful method for use with adaptive optimizers for improving the performance by overcoming loss flattening and loss of adaptivity. In §4 we conduct experiments on image classification and recommender datasets and show that Hoffer et al's method is quite inferior to LAWN.

---

[4]In fully homogeneous nets, layer-wise weight normalization is a way of removing some redundancies (Dinh et al., 2017); layer-wise weight norms also connect well with implicit bias properties (Lyu & Li, 2020).

**LAWN Implementation.** There are two ways of implementing the constrained phase of LAWN. The first method is to define an unconstrained vector $v^\ell$ and set $w^\ell = c^\ell \, v^\ell / ||v^\ell||$ and simply apply the optimizer to $v^\ell$. This is the implementation suggested by Salimans and Kingma (Salimans & Kingma, 2016); see equation (2) there. (Note, however, that Salimans and Kingma (Salimans & Kingma, 2016) keep the radial component by including the scale parameter, $g$.) The downside with this method is that it puts load on the computational graph, and automatic differentiation through the normalizing transformation increases the computational cost (Huang et al., 2020).

The second method is to have the optimizer directly deal with the constraints. At a given weight vector, $w$, let $g = \nabla_w L(w)$. For updating $w^\ell$, the projected gradient defined by

$$g_p^\ell = g^\ell - \frac{(w^\ell)^T g^\ell}{\|w^\ell\|^2} w^\ell \quad \forall \ell \tag{3}$$

naturally plays the role of the gradient for decreasing the loss on the manifold defined by $\|w^\ell\| = c^\ell$ and so it is used instead of the gradient in all optimizer related updates. In Appendix D we use gradient flow to establish this. We use this method in our implementation of the constrained phase of LAWN. The implementation of the constrained phase of Adam-LAWN is given in Algorithm 1.

Let us now describe the LAWN method as a complete algorithm. The free and constrained training of LAWN can be done with any given optimizer. LAWN does a total of $E_{total}$ epochs; $E_{total}$ is fixed for a given dataset. After $E_{free}$ epochs of free training, with $\bar{w}$ denoting the weights reached, it sets $c^\ell = \|\bar{w}\|^\ell \; \forall \ell$ and switches to do constrained training (solve (2)) for the remaining $(E_{total} - E_{free})$ epochs. Learning rate schedules are important for deep networks to attain good performance and LAWN employs the standard linear warmup and decay schedule (Loshchilov & Hutter, 2019; Liu et al., 2020; Loshchilov & Hutter, 2016). This schedule has two hyperparameters, $\eta_{peak}$, the peak learning rate, and $E_{warmup}$, the number of warmup epochs. $E_{free}$ is the additional hyperparameter; as already mentioned, this hyperparameter replaces the weight decay parameter.

**LAWN and large batch sizes.** For a fixed epoch budget, larger batch sizes require a smaller number of steps; combined with distributed computation this helps speed up training. However, since stochasticity of updates reduces with large batch size, the mechanism of escape from sub-optimal solutions gets affected. Thus, one usually sees a reduction in generalization performance as batch size is increased (Shallue et al., 2018). Loss flattening makes this issue worse by affecting adaptivity. LAWN, by helping overcome this issue, leads to a more graceful degradation of performance as a function of batch size. We will empirically demonstrate this in §4. The degradation becomes far less (even zero) when the the total number of steps is allowed to decently increase with batch size.

**LAWN and weight adaptivity.** Research in the last five years is clearly showing that improving weight adaptivity is the key to escaping inferior weights and reaching weights with superior generalization performance. It was believed that the noise associated with stochastic gradient is the only way to ensure such adaptivity, leading to the promotion of smaller batch sizes and larger learning rates. But other ways of improving adaptivity are being suggested. For example, injection of suitable forms of artificial noise has been shown to improve generalization (Wu et al., 2020; Wen et al., 2020). It is even being suggested that adaptivity can be improved without any stochasticity and with just suitable deterministic regularization (Geiping et al., 2021). More interesting research is expected on this important topic of weight adaptivity. In this line of research and methods, LAWN can be thought of as an orthogonal technique of improving adaptivity by suitably constraining the norms of the weights and avoiding loss flattening. While weight decay also helps in a similar way, we will show in §4 that LAWN is much superior.

## 4 EXPERIMENTS

Our baselines include SGD, Adam (Kingma & Ba, 2017) and LAMB (You et al., 2020). We add weight decay to all 3 baseline algorithms, and additionally add momentum to SGD. To evaluate LAWN, we consider LAWN-based variants of the adaptive optimizers (Adam and LAMB). SGD has demonstrated strong performance for computer vision tasks (Ren et al., 2015; Goyal et al., 2018), whereas adaptive methods like Adam perform well on other domains (eg. recommender systems, text classification). To demonstrate the efficacy of LAWN across a wide variety of tasks, we conducted experiments on the CIFAR (Krizhevsky et al., 2009) and ImageNet (Deng et al., 2009; Krizhevsky et al., 2012) datasets for image classification, and the MovieLens (Harper & Konstan, 2015) and Pinterest (Geng et al., 2015) datasets for item recommendation. All model training code

was implemented using the PyTorch library (Paszke et al., 2019) and experiments were conducted on machines with NVIDIA V100 GPUs. For each experiment, we report average test metric over 3 runs.

**Hyperparameters:** Regular versions of SGD, Adam, and LAMB use weight decay. The LAWN versions do not use weight decay; instead, they use $E_{free}$. For a given dataset, the total number of epochs, $E_{total}$ was fixed. Learning rate schedule (with warmup of learning rate from 0 to $\eta_{peak}$ in $E_{warmup}$ epochs followed by decay to zero in the remaining epochs) has proved beneficial. The above mentioned hyperparameters were all tuned to get the best generalization performance. For $\eta_{peak}$ we used equally spaced values in logarithmic scale suited for each (optimizer, dataset) combination. The value of $E_{total}$, the range of values for weight decay, $E_{free}$ and $E_{warmup}$, and additional details used for individual datasets are given in Appendix B. Apart from these, SGD's momentum value was fixed at 0.9, unless specified otherwise in Appendix B. For Adam, LAMB, Adam-LAWN and LAMB-LAWN, we fixed $\beta_1 = 0.9$ and $\beta_2 = 0.999$. For Adam, we used $\epsilon = 10^{-8}$ and the rest of the adaptive optimizers use $10^{-6}$. We did not tune $\epsilon$, $\beta_1$ and $\beta_2$, which could have led to further improvements.

Details about datasets, pre-processing and network architectures for all experiments can be found in Appendix B.

### 4.1 LAWN VS. OTHER METHODS FOR CONTROLLING LOSS OF ADAPTIVITY

| Method | MovieLens-1M | | CIFAR-10 | | CIFAR-100 | |
|---|---|---|---|---|---|---|
| | $BS = 10k$ | $BS = 100k$ | $BS = 4k$ | $BS = 10k$ | $BS = 4k$ | $BS = 10k$ |
| LSR | 68.66 | 67.34 | 93.09 | 92.73 | 69.92 | 69.24 |
| WD | 70.12 | 69.28 | 92.93 | 92.63 | 68.91 | 68.61 |
| Hoffer | 44.54 | 45.52 | 92.99 | 91.66 | 70.66 | 69.35 |
| LAWN | **70.41** | **70.77** | **93.74** | **93.84** | **73.13** | **72.97** |

**Table 2:** Comparison of test performance of LAWN with other methods for controlling loss of adaptivity on Movielens-1M, CIFAR-10 and CIFAR-100 datasets. The base optimizer used is Adam. Two different batch sizes, $BS$ are tried for each dataset. For LSR, the smoothing parameter was fixed at 0.05. LAWN comprehensively outperforms other methods, including weight decay (WD).

We first compared LAWN to three key methods for controlling loss of adaptivity, discussed in §2.3. The comparison is done on the three datasets, Movielens-1M, CIFAR-10 and CIFAR-100. For each dataset we tried two values of batch size. Table 2 gives the results. LAWN clearly outperforms all the other methods. The second overall best method is weight decay and hence it is used as the baseline for all remaining experiments of this section. The weight normalization technique suggested by Hoffer et al (Hoffer et al., 2018) does very badly on Movielens-1M; on CIFAR-10 and CIFAR-100, though it gives a decent performance, it lags behind LAWN a lot. Clearly, this method requires a modification along the lines of LAWN in order for it do well.

### 4.2 IMAGE CLASSIFICATION FOR CIFAR-10 AND CIFAR-100

For both CIFAR-10 and CIFAR-100 (Krizhevsky et al., 2009), we used the VGG-19 CNN network (Simonyan & Zisserman, 2014) with 1 fully connected final layer. Our ImageNet experiments use a ResNet-based (He et al., 2015) architecture. All experiments were run with a 300 epoch budget. As seen in Table 3, LAWN variants either match or outperform the base variants across batch sizes. Adam-LAWN is particularly impressive. This is in stark contrast to earlier held beliefs that adaptive optimizers cannot match SGD's generalization performance for image classification tasks (Wilson et al., 2017).

**Effect of batch size.** LAWN variants cause more graceful degradation of performance with batch size, as compared to base variants. Adam-LAWN causes almost no degradation in generalization performance even at batch size 10k (see Figures 2(a), 2(b)).

**Effect of $E_{free}$.** We observed that switching early to LAWN mode (i.e. fixing $E_{free}$ to less than 10 epochs) usually works well for generalization. See Appendix B for details. This is consistent with our hypothesis that constrained training should kick in before loss flattening sets in.

### 4.3 RECOMMENDATION SYSTEMS

We conducted experiments on the MovieLens-100k, MovieLens-1M and Pinterest datasets using the popular neural collaborative filtering (NCF) technique (He et al., 2017). The datasets contain ratings provided to various items by users. The model is a 3-layer MLP, the input being user and item embeddings. We use the *hit ratio@10* metric (expressed as %), where we reward the model for

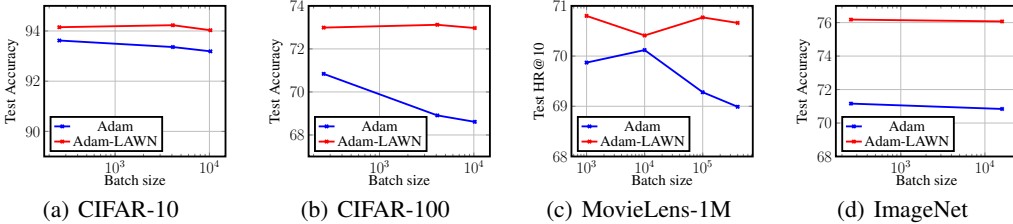

**Figure 2:** Adam-LAWN vs. Adam (weight decay comprehensively tuned) for a variety of datasets. Adam-LAWN causes little to no drop in generalization performance with increasing batch size.

ranking a test item in the top 10 of 100 randomly sampled items that a user has not interacted with in the past. We trained for 300 epochs for the two smallest batch sizes, and 500 epochs for the two biggest batch sizes for each dataset. Details about the datasets, pre-processing, model and evaluation can be found in Appendix B. A summary of the performance of the aforementioned optimizers on all 3 datasets can be found in Table 4. LAWN-based optimizers consistently outperform their base variants. SGD failed to generalize well at large batch sizes and this requires further investigation.

| Method | MovieLens-100k | | | | MovieLens-1M | | | | Pinterest | | | |
|---|---|---|---|---|---|---|---|---|---|---|---|---|
| | 1k | 10k | 100k | 400k | 1k | 10k | 100k | 1M | 1k | 10k | 100k | 1M |
| SGD | 66.33 | 65.58 | Fail | Fail | 70.91 | 69.31 | Fail | Fail | 86.62 | 85.57 | Fail | Fail |
| Adam | 66.01 | 66.03 | 63.20 | 63.98 | 69.87 | 70.12 | 69.28 | 68.99 | **87.27** | 85.97 | 85.81 | 85.30 |
| Adam-L | **66.81** | **66.91** | **66.24** | **66.14** | **70.80** | **70.41** | **70.77** | **70.66** | 86.85 | **86.61** | **86.04** | **86.06** |
| LAMB | 65.45 | 65.34 | 64.23 | 62.57 | 69.91 | 69.77 | 69.44 | 68.95 | 86.63 | 85.91 | 85.80 | 85.65 |
| LAMB-L | **66.56** | **66.54** | **66.52** | **66.14** | **70.86** | **70.86** | **70.68** | **70.34** | **86.83** | **86.25** | **85.99** | **86.07** |

**Table 4:** Test HR@10 on MovieLens and Pinterest recommendations. Standard error is in the range $[0.15, 0.25]$; details are in Appendix B. Highlighted values indicate the better performing method between x and x-L.

**Weight decay vs. LAWN.** Weight decay was used and tuned for all the base optimizers since it arrests the uncontrolled growth of network weights, helping avoid of loss of adaptivity. The LAWN variants do not use weight decay but still outperform the base variants.

**Effect of batch size.** LAWN variants of Adam and LAMB scale to very large batch sizes (1 million for MovieLens-1M, 400k for MovieLens-100k) without any appreciable loss in accuracy. SGD could only scale to batch size

| Method | CIFAR-10 | | | CIFAR-100 | | |
|---|---|---|---|---|---|---|
| | 256 | 4k | 10k | 256 | 4k | 10k |
| SGD | 93.99 | 93.48 | 92.99 | 73.49 | 71.68 | 71.07 |
| Adam | 93.48 | 92.93 | 92.63 | 70.84 | 68.91 | 68.61 |
| Adam-L | **93.91** | **93.74** | **93.84** | **72.99** | **73.12** | **72.97** |
| LAMB | **93.76** | **93.27** | 92.91 | **71.29** | 69.39 | 67.76 |
| LAMB-L | 93.67 | 93.22 | **92.92** | 71.25 | **69.68** | **69.16** |

**Table 3:** Test accuracy on CIFAR-10 and CIFAR-100. Standard error is in the range $[0.1, 0.45]$. Details are in Appendix B. Highlighted values indicate the better performing method between x and x-L.

10k. Adam-LAWN's strong scalability with batch size is consistent with results obtained from the CIFAR experiments (also see Figure 2(c)).

**Effect of $E_{free}$.** Similar to the results of the CIFAR experiments, fixing $E_{free}$ to a small value works well for LAWN. Details are in Appendix B.

## 4.4 IMAGE CLASSIFICATION FOR IMAGENET

As compared to CIFAR, the ImageNet classification problem (Krizhevsky et al., 2012) is more representative of real world classification problems. We used a variant of the popular ResNet50 (He et al., 2015) model as the classifier. We considered a small (256) and a large (16k) batch size for this experiment, and fixed training budget to be 90 epochs.

**Results for batch size 256.** Overall results can be found in Table 1 (see §1). SGD, used in conjunction with momentum and weight decay, has long been the optimizer of choice for image classification. Adam is well known to perform worse than SGD for image classification tasks (Wilson et al., 2017). For our experiment, we tuned the learning rate and could only get an accuracy of 71.16%. In comparison, Adam-LAWN achieves an accuracy of more than 76%, marginally surpassing the performance of SGD.

We found it difficult to reproduce ImageNet results using the LAMB algorithm. We made minor modifications (details in Appendix B) to the original algorithm to make it more stable, and call

the resultant algorithm LAMB+. LAMB-LAWN (the LAWN version of the unmodified LAMB) comprehensively outperforms LAMB+ for batch size 256 by achieving an accuracy close to 76.5%.

**Results for batch size 16k.** For the large batch size of 16k, we noticed that LAWN retains strong generalization performance (also see Figure 2(d)). Both Adam-LAWN and LAMB-LAWN achieve very high accuracy, with Adam-LAWN retaining its performance at such a large batch size by crossing the 76% test accuracy mark. This is with only additonally tuning for the LAWN variants $E_{free}$ and $E_{warmup}$.

**Remark.** Adam has traditionally performed worse than SGD at tasks like image classification. Recent work (Choi et al., 2020; Nado et al., 2021) has shown that Adam's inner hyperparameters (that include $\beta_1, \beta_2, \epsilon$) could be the reason for the inferior generalization. The above cited works use sophisticated hyperparameter tuning algorithms over a relatively large search space (see Appendix D of Choi et al. (2020)) and conclude that the optimal parameters vary a lot between datasets. While these are important results to close the gap in our understanding of Adam, they do little to improve the practical usability of Adam since it is prohibitively expensive to run the recommended number of training runs required to find the ideal hyperparameters.

### 4.5 LAWN WORKS WITH OTHER LOSS FUNCTIONS

To understand the effect of LAWN on loss functions other than cross entropy, we conducted experiments with focal loss (Lin et al., 2017). Focal loss was proposed as a way of improving performance when the classification problem is highly imbalanced, and has also recently found use for calibration of neural networks (Mukhoti et al., 2020). It is very different from cross-entropy, but still suffers from the issue of loss flattening. We conducted experiments using focal loss and Adam on both item recommendation and image classification. Results are given in Table 5. Adam-LAWN outperformed regular Adam with weight decay on each one of the cases. This demonstrates the strengths of LAWN across a variety of loss functions and reinforces LAWN's efficacy in improving generalization performance.

| Method | MovieLens-1M | | | | CIFAR-10 | | | | CIFAR-100 | | | |
|---|---|---|---|---|---|---|---|---|---|---|---|---|
| | $FL_{0.5}$ | $FL_2$ | $FL_5$ | CE | $FL_{0.5}$ | $FL_2$ | $FL_5$ | CE | $FL_{0.5}$ | $FL_2$ | $FL_5$ | CE |
| Adam (BS1) | 67.33 | 69.02 | 67.17 | 69.28 | 93.07 | 91.16 | 87.88 | 92.93 | 69.19 | 68.85 | 67.72 | 68.91 |
| Adam-L (BS1) | **71.24** | **69.68** | **69.24** | **70.77** | **93.89** | **92.22** | **89.14** | **93.74** | **73.05** | **73.29** | **72.89** | **73.12** |
| Adam (BS2) | 66.42 | 67.40 | 66.62 | 68.99 | 92.51 | 89.79 | 86.82 | 92.63 | 67.94 | 67.28 | 66.11 | 68.61 |
| Adam-L (BS2) | **70.53** | **69.56** | **68.66** | **70.66** | **93.21** | **91.20** | **88.93** | **93.84** | **72.51** | **72.55** | **70.95** | **72.97** |

**Table 5:** Adam vs. Adam-LAWN when used with focal loss (FL). We tried 3 different values (0.5, 2 and 5) for $\gamma$ (focal loss parameter) and also compared the results to cross entropy (CE) loss. BS1 and BS2 refer to batch sizes. For MovieLens-1M, BS1 = 100k and BS2 = 1M. For the CIFAR datasets, BS1 = 4k and BS2 = 10k.

### 4.6 LAWN IMPROVES CALIBRATION

| Method | CIFAR-10 | CIFAR-100 |
|---|---|---|
| Adam | 0.132 | 0.292 |
| Adam-L | **0.048** | **0.188** |

**Table 6:** ECE (lower is better) on image classification.

A well calibrated network is one in which the predicted class probability is close to the observed probability of being correct. Many applications require a well calibrated network. It is well known that over-parameterized deep nets are prone to over predicting probabilities (Guo et al., 2017). One of the main reasons for this is that weights become large, logits become large, and so the network is pushed to give out extreme probability values. Weight decay helps Adam improve calibration over unregularized Adam. Here we demonstrate that Adam-LAWN significantly improves the calibration even over Adam with weight decay. Estimated Calibration Error (ECE) is a standard metric for measuring calibration error (Guo et al., 2017). For CIFAR-10 and CIFAR-100, with all hyperparameters tuned, Table 6 gives the ECE values for Adam and Adam-LAWN. It is clear that Adam-LAWN gives much smaller ECE values than Adam (with weight decay). This improved calibration is an important advantage of LAWN.

## 5 CONCLUSION

LAWN as a simple and powerful method of modifying deep net training with a base optimizer to improve weight adaptivity and lead to improved generalization. Switching from free to weight norm constrained training at an appropriate point is a key element of the method. We study the performance of the LAWN technique on a variety of tasks, optimizers and batch sizes, demonstrating its efficacy. Tremendous overall enhancement of Adam and the improvement of all base optimizers at large batch sizes using LAWN are important highlights.

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
