# OpenReview forum: "Logit Attenuating Weight Normalization"
_ICLR.cc/2022/Conference — ICLR 2022 Submitted_

### Official Review · Reviewer_qsYk · 2021-10-25

**Correctness:** 3
**Technical Novelty And Significance:** 2
**Empirical Novelty And Significance:** 3
**Recommendation:** 5
**Confidence:** 4

**Main Review:**

# Strengths:

The motivation for this work is straightforward and reasonable. In a homogeneous neural network, the prediction in classification tasks is invariant to the scale of logits and weights. However, as the neural network is over-parameterized it's often fitting the training data quite well and the weight scales are increased to reduce the loss. The paper argues that the loss flattening worsens the generalization since the model cannot escape from local minimum easily. The proposed LAWN algorithm is clearly presented and some hyperparameter settings are given in the appendix.


# Weakness:

My major concern for this work is that it neglects batch normalization (BN) layers which are crucial to the performance of ResNets but are invariant to weight norm scaling. In other words, the loss flattening problem cannot be alleviated by constraining weight norms in networks with BN because scaling weight norms in BN has no impact on the logit scales. If the paper argues that even in networks with BN the LAWN still has its advantage, this argument should be explicitly made in the paper. Though a ResNet50 is used in the ImageNet experiment, I did not find any details on how to deal with BN.

Another concern is about the implementation of weight norm constraint. If my understanding is correct, Equation (3) is how the constraint is enforced. Appendix D shows the projected gradient will not increase the weight norm assuming the learning rate is infinitesimal. However, a large learning rate is necessary for the generalization of DNNs and is often used in training practice of DNNs. In fact, the projected gradient always increases the norm with a practical learning rate since it is orthogonal to the weight, and the learning rate controls the increase of the weight norm. Compared with weight normalization (Salimans & Kingma 2016), Equation (3) is the gradient of v^l when c^l is 1. The difference is not so clear from my point of view.

Overall, I think the paper's analysis on the loss flattening is interesting and I appreciate the effort of authors. But the current manuscript is not ready to be published at ICLR. I hope the paper can be revised to address my above concerns.


**Summary Of The Paper:**

The paper proposes to overcome the loss flattening problem in deep neural networks by explicitly constraining the weight norm during training. The paper argues that as a result of scale-variant loss functions in classification and ranking tasks, the loss can be made small by pushing logits to large magnitude. This loss adaptivity phenomenon is argued to hurt generalization since the escaping condition in SGD training is difficult to satisfy if the logits are large. To alleviate such a problem, the paper proposes to fix weight norms during training (LAWN), which is implemented by projected gradient. The empirical result on several datasets shows the advantage of LAWN, especially when a large batch size is used.

**Summary Of The Review:**

In spite of its interesting motivation, the paper does not properly handle DNNs with Batch Norm such as ResNets and many variants, and the  way the weight norm constraint is implemented is confusing.

---

> ### Author Response · Authors · 2021-11-23
> **Initial response to reviewer qsYk**
>
> We thank the reviewer for the various positive comments about the paper.
>
> Currently, we have not made any changes to the submitted paper. We will make the necessary changes after discussion with the reviewers in the rebuttal phase.
>
> Though mentioned in our submitted paper, we want to quickly highlight the following important and crucial contributions of this paper for deep net optimization.
> 1. Loss flattening and the resulting inability to escape from poor minima are key reasons for loss of performance.
> 2. Though the importance of the containment of weight norms has been pointed out in previous works, LAWN gives a simple, efficient and effective way to set the norm values to achieve high performance.
>
> **Response to the reviewer’s comments in the Main Review**
>
> 1. On batch normalization and invariance to weight scaling. From a forward propagation point of view, the reviewer is very right. However, from the point of view of gradient calculation and the effect on the step sizes (and hence the steps taken and the progress of the training algorithm), weight normalization will make a good difference. This is discussed in detail in [1] and [2].  We use BN for both the VGG19 and ResNet50 models in our paper. We agree with the reviewer that a discussion on why the normalization of weights in BN is helpful needs to be added to the paper.
>
> 2. On the enforcing of the weight norm constraint: In step 11 of Algorithm 1 (Adam-LAWN Constrained phase) we explicitly rescale the weights so that weight norm constraints are satisfied. So eq(3) holds. Since this step is hidden inside the algorithm, this is something that we will make more clear where we describe the method in the main text.
>
> 3. Compared with weight normalization (Salimans & Kingma 2016), Equation (3) is the gradient of $v^\ell$ when $c^\ell$ is 1. The difference is not so clear from my point of view. The reviewer is correct in making this observation. Like we mentioned at the top of page 6, both methods - (a) working with gradient projection and (b) using unconstrained variables, $v^\ell$ as in Salimans and Kingma 2016 -  are good for implementing LAWN. We simply preferred working with method (a) so as to avoid putting a load on the computational graph.
>
> **Reference**
> 1. Heo B, Chun S, Oh S J, et al. AdamP: Slowing down the slowdown for momentum optimizers on scale-invariant weights[J]. arXiv preprint arXiv:2006.08217, 2020.
> 2. Elad Hoffer, Ron Banner, Itay Golan, and Daniel Soudry. Norm matters: efficient and accurate normalization schemes in deep networks. In Advances in Neural Information Processing Systems, 2018.

---

### Official Review · Reviewer_7vWF · 2021-11-01

**Correctness:** 3
**Technical Novelty And Significance:** 3
**Empirical Novelty And Significance:** 3
**Recommendation:** 5
**Confidence:** 4

**Main Review:**

STRENGTHS:
- The results are promising and the paper contains a lot of experiments. However, some of them are very specific to recommender systems. (See below.)
- The LAWN modification seems to improve the performance of adaptive optimizers on image-classification and recommendation tasks as well as the calibration of deep neural network classifiers.
- The author(s) is/are very transparent about the limits of their method's applicability.

WEAKNESSES:
- Although the author(s) state in the abstract that they want to address the generalization gap between adaptive-optimizers and SGD in the image classification domain, most experiments are limited to very specific recommender systems datasets. The only image-classification datasets considered in this work are CIFAR-10, CIFAR-100 and Imagenet, which limits my enthusiasm for this paper.
- There are some issues with the notation: using $nc$ to denote the number of classes is problematic. I would suggest to use subscripts (e.g., $n_c$ for number of classes or $n_w$ for the number of weight variables) instead of variable names consisting of two letters. Equation 3 defines $g_p^\ell$, however in Algorithm 1 $g_{pt}^\ell$ (line 4) is used. Perhaps it would be better to define Equation 3 also depending on some time step t to make clear that this is what line 4 of Algorithm 1 is referring to.
- I find it strange that SGD failed in the MovieLens and Pinterest datasets for very large batch-sizes. As a reader i am curious why and how it fails. It would be nice if the author(s) elaborate(s) on that matter during the rebuttal period. What accuracy does SGD achieve in that case?
- Why is it necessary to start training the network in a free (unregularized) mode? What happens when the $c^\ell$ is constant? Does the value for $c^\ell$ really differ for each dataset/model? Which test accuracy does one get when setting $E_\text{free}$ to 0 epochs, i.e., when using the norm of the initial weight vectors for normalization? At least for larger batch sizes (4k or 10k) the trend shown in the Figure 4 of the supplementary material seems to suggest that this could further improve the accuracy.
- Table 2 could be extended to also include similar experiments on the Imagenet dataset.

MINOR REMARKS:
- Multiple wrong citations, e.g., "Hoffer et al (Hoffer et al., 2018)" instead of "Hoffer et al. (2018)" or "Salimans and Kingma (Salimans & Kingma, 2016)" instead of "Salimans and Kingma (2016)"

**Summary Of The Paper:**

The paper proposes a technique called Logit Attenuating Weight Normalization (LAWN) which is a modification that can be applied to gradient-based optimizers. LAWN combines a gradient projection step with weight normalization to overcome the short-comings of adaptive gradient-based optimizers especially when being applied to large batch sizes. The LAWN method is applied to different deep learning models for image-classification and movie recommendations and increased performance is demonstrated. It is also shown that LAWN improves network calibration.

**Summary Of The Review:**

Overall, see some issues with this submission that could be addressed by the author(s) during the rebuttal. I am willing to improve my score when my concerns are addressed appropriately.

---

> ### Author Response · Authors · 2021-11-23
> **Initial response to reviewer 7vWF**
>
> We thank the reviewer for the various positive comments about the paper.
>
> Currently, we have not made any changes to the submitted paper. We will make the necessary changes after discussion with the reviewers in the rebuttal phase.
>
> Though mentioned in our submitted paper, we want to quickly highlight the following important and crucial contributions of this paper for deep net optimization.
> 1. Loss flattening and the resulting inability to escape from poor minima are key reasons for loss of performance.
> 2. Though the importance of the containment of weight norms has been pointed out in previous works, LAWN gives a simple, efficient and effective way to set the norm values to achieve high performance.
>
> **Response to the reviewer’s comments in the Main Review**
>
> 1. On doing more image datasets: Most established papers on deep net optimizers report experimental results only on CIFAR-10, CIFAR-100 and ImageNet. In particular, showing good improvements on ImageNet is considered as the key test of a method. If the reviewer can point to a specific dataset, then we can do more experiments and report results.
>
> 2. On some issues with the notations: Yes, we will replace $nc$ with $n_c$, etc. In Algorithm 1, $g_{pt}^\ell$ had the additional t in the subscript to denote the step number. We can remove the t if it causes confusion.
>
> 3. On the failure of SGD in the Movielens and Pinterest datasets: By the word “failure”, we meant that SGD achieves poor accuracy (despite the training loss becoming low) for large batch sizes. For instance, for a batch size of 1M, SGD on the MovieLens-1M dataset yields 45 as the hit ratio @10. This is significantly lower than the numbers achieved using other optimizers.
>
> 4. On the importance of the free phase: The initial phase of deep net training is crucial because the various parts of the network set themselves rapidly to their main feature functions in this phase. The generalization performance rises sharply in this phase. There is no need for weight decay or logit containment for Adam in this stage; in fact, they are hindrances in this phase. It is only later when the various parts of the network start forming more intricate feature formations that weight decay or logit containment needs to be introduced. The right length of the free phase for achieving the best final performance is dependent on the dataset. Hence we use $E_{\text{free}}$ as a hyperparameter and tune it. This automatically sets the right values for the $c^\ell$. Yes, the value for $c^\ell$ differs a lot for each dataset/model. We can expand on this discussion in the paper.
>
> 5. On setting $E_{\text{free}}$ even below one epoch or even zero epochs. We want to point out (also see our response to item 4 above) that the free phase is important. Hence, setting $E_{\text{free}}$ to zero leads to a significant loss in performance. This is also the reason why Huber et al’s method [2] and AdamP [1] (please see our response to Reviewer Xf6G for a more detailed discussion of AdamP) do not do as well as LAWN. If Figure 4 is extended to values of $E_{\text{free}}$ below one epoch, it would dip in performance at low values. But we agree that the best value of $E_{\text{free}}$ could be smaller than one.
>
> 6. Table 2 could be extended to also include similar experiments on the Imagenet dataset. Due to time and resource constraints, we are unable to conduct experiments on the ImageNet datasets. However, we believe that the results on the CIFAR datasets are indicative and representative of classification performance on ImageNet.
>
> 7. When we rewrite the paper we will address the minor remark on doing citations correctly.
>
> **Reference**
>
> 1. Heo B, Chun S, Oh S J, et al. AdamP: Slowing down the slowdown for momentum optimizers on scale-invariant weights[J]. arXiv preprint arXiv:2006.08217, 2020.
> 2. Elad Hoffer, Ron Banner, Itay Golan, and Daniel Soudry. Norm matters: efficient and accurate normalization schemes in deep networks. In Advances in Neural Information Processing Systems, 2018.

---

### Official Review · Reviewer_dDDW · 2021-11-02

**Correctness:** 3
**Technical Novelty And Significance:** 2
**Empirical Novelty And Significance:** 3
**Recommendation:** 5
**Confidence:** 3

**Main Review:**

Overall, I appreciate the simple but effective weight normalization method for network training. My major concerns are on the reason/analysis of the effectiveness of LAWN, the sensitivities to its hyper-parameters, the clarity on the details of the model, and comparisons.

1.  It is understandable that the weight scale should be bounded to improve the stability of network prediction. However, the reason of why the proposed normalization in Eqn.(2) is effective compared with the other approaches, such as the weight decay, LSR, etc., should be more deeply analyzed.

2.  For Eqn. (3), according to appendix D, it is to implement the constrain of lines 4 and 9 in algorithms 1, for optimizing eqn.(2) under the weight constraints. I am afraid that the gradient descent in eqn.(3) may not exactly find the projected gradient, can it practically guarantee that the weights constraint can be satisfied when optimizing eqn. (2)?

3. The sensitivity of performance to parameter c^l in Eqn. (2) should be given in details. Though it is heuristically set as |\bar{w}^l|, whether the proposed algorithm is sensitive to this parameter should be analyzed.

4. The proposed algorithm is mainly compared with the baseline algorithms such as LSR, weight decay. There are several related works on improving generalization of training algorithm by escaping the bad local minima. The authors are suggested to more thoroughly compare with more related state-of-the-art training algorithms in literature.

**Summary Of The Paper:**

This paper proposed a network training algorithm to overcome the bad local minima for better network generalization. The proposed method, dubbed logit attenuating weight normalization (LAWN) is to constrain the weight norms to constrain the value of logit / softmax. The experiments justifies that the proposed method can be easily combined with different optimizers and improved the generalization ability of learned network.

**Summary Of The Review:**

The proposed training algorithm is simple, but showed to be able to improve network generalization ability apparently.  I mostly like the results, but the proposed algorithm should be more thoroughly analyzed and compared as pointed out in the above comments.

---

> ### Author Response · Authors · 2021-11-23
> **Initial response to reviewer dDDW**
>
> Currently, we have not made any changes to the submitted paper. We will make the necessary changes after discussion with the reviewers in the rebuttal phase.
>
> Though mentioned in our submitted paper, we want to quickly highlight the following important and crucial contributions of this paper for deep net optimization.
> 1. Loss flattening and the resulting inability to escape from poor minima are key reasons for loss of performance in deep nets.
> 2. Though the importance of the containment of weight norms has been pointed out in previous works, LAWN gives a simple, efficient and effective way to set the norm values to achieve high performance.
>
> **Response to the reviewer’s comments in the Main Review**
>
> 1. On the reason why the proposed normalization in Eqn.(2) is effective compared with the other approaches, such as the weight decay, LSR, etc. Let us begin with weight decay. The trajectories of weight decay move freely in the weight space, though they are pressured to reduce the weight sizes. On the other hand, constraining weight norms, as in LAWN, directly controls the size of the logits and hence the weight adaptivity and the ability to escape from inferior minima. Also, in LAWN, different layers use different norm values, which is akin to regularizing different layers differently. These different norm values for different layers, $c^\ell$, are naturally determined by the free-to-constrained switch point ($E_{\text{free}}$). Weight decay on the other hand uses a single regularization constant for all layers. Note that both methods have only one additional hyper-parameter to tune - either $E_{\text{free}}$ or weight decay coefficient.  Let us take LSR. It is well known, right from the work [1] that introduced the idea, that LSR alone is not a powerful regularizer by itself and it is not sufficient to obtain great performance. With over-parameterized networks, LSR avoids a loss flattening behavior, which is good, but the loss (KL divergence) itself gravitates towards zero easily.
>
> 2. Satisfying the weight norm constraints. In step 11 of Algorithm 1 (Adam-LAWN Constrained phase) we explicitly rescale the weights so that weight norm constraints are satisfied. So eq (3) holds. Since this step is hidden inside the algorithm, this is something that we will make more clear where we describe the method in the main text.
>
> 3. The sensitivity of performance to parameter c^l in Eqn. (2) should be given in detail. Though it is heuristically set as |\bar{w}^l|, whether the proposed algorithm is sensitive to this parameter should be analyzed. It is important to note that the $c^\ell$ are not hyperparameters under our control. They are automatically decided by the free phase of LAWN and the single hyperparameter, $E_{\text{free}}$. Thus, instead of discussing sensitivity with respect to the $c^\ell$, it is better to talk of sensitivity with respect to $E_{\text{free}}$. Though the best choice of $E_{\text{free}}$ depends on the dataset, a small number of epochs (10% or less of the total number of epochs) is usually sufficient. Also, it is not necessary to do a fine search of $E_{\text{free}}$ to get the best performance. See Appendix B of the supplement for some experiments on the effect of $E_{\text{free}}$.
>
> 4. The proposed algorithm is mainly compared with the baseline algorithms such as LSR, weight decay. There are several related works on improving the generalization of training algorithms by escaping the bad local minima. The authors are suggested to more thoroughly compare with more related state-of-the-art training algorithms in the literature. AdamW is one of the most popularly used methods and our aim in this paper is to improve it. The best recent methods for escaping bad (sharp) local minima - for example, SAM, which is based on adversarial training ideas, and Apollo, which is based on second-order training, are based on ideas that are orthogonal to LAWN. So, there is even the possibility of combining them with LAWN. This, as well as a thorough evaluation of LAWN against those best recent methods, is of a much larger scope and will be taken up in follow-up work.
>
> **Reference**
> 1. Christian Szegedy, Vincent Vanhoucke, Sergey Ioffe, Jonathon Shlens, and Zbigniew Wojna. Rethinking the inception architecture for computer vision. CoRR, abs/1512.00567, 2015.

---

### Official Review · Reviewer_Xf6G · 2021-11-04

**Correctness:** 3
**Technical Novelty And Significance:** 2
**Empirical Novelty And Significance:** 2
**Recommendation:** 3
**Confidence:** 5

**Main Review:**

1. The proposed method is almost the same as AdamP [1]. The difference is there are two gradient projection operation.
    The motivation and main approach is very similar to AdamP. But the author did not compare with AdamP and even not mentioned it in
    related works.
2. The compared method Adam is not fair. It is well-known that Adam with L2 regularization weight decay leads to bad generalization.
    A proper way to add weight decay is to use the decoupled weight decay, i.e., AdamW. Although the author mentioned AdamW in related
   work section, it seems not be a comparison method in the experimental part.
3.  Please give the illustration of how to tune E_{free} for different datasets and different learning processes.
4.  Actually, only using Eq (3) on gradient can not make sure Eq (2) hold. The norm of the weight will still gain. There should be a weight
    projection step. But I did not find it in the proposed method. Why do not use weight  projection step to keep the norm of weight
    unchanged? If the norm of the weight changes in each step, the author should not state that  the proposed method optimizes the
    constrained objective function in Eq (2).






[1] Heo B, Chun S, Oh S J, et al. AdamP: Slowing down the slowdown for momentum optimizers on scale-invariant weights[J]. arXiv preprint arXiv:2006.08217, 2020.



**Summary Of The Paper:**

The author propose a new regularization method for training the deep neural networks, instead of weight decay.
The proposed method is named Logit Attenuating Weight Normalization (LAWN), which constrains the weight norm in
training with the projected gradient. The experimental results show LAWN is more effective than weight decay in Adam and
LAMB optimizers.

**Summary Of The Review:**

The similar idea has been conducted in the work AdamP. And the compared methods are not convincing in the experiment part.

---

> ### Author Response · Authors · 2021-11-23
> **Part 1 of response to reviewer Xf6G**
>
> Currently we have not made any changes to the submitted paper. We will make the necessary changes after discussion with the reviewers in the rebuttal phase.
>
> We thank the reviewer for pointing out the AdamP paper [1]. At the same time, we feel sad to see the reviewer raising an “ethics concern”. The AdamP paper is not cited in the mainstream papers on deep learning optimizers and that is the reason we missed it. It was certainly not a deliberate omission.
>
> We also wish to point out that, in terms of the method as well as performance, AdamP is close to Hoffer et al’s method [2], with which we have compared LAWN and shown that LAWN has superior performance; details are in our paper.
>
> As a part of this rebuttal, below we compare Adam-LAWN with AdamP and show that Adam-LAWN has superior performance, especially at large batch sizes.
>
> We will include a discussion of AdamP and these comparison results in the main paper after discussion with the reviewer.
>
> **Response to the reviewer’s four comments in the Main Review**
>
> 1. Drawbacks with AdamP
>
> We agree that AdamP (which refers to both the paper as well as the algorithm) clearly points out the importance of containing weight norms. However, it has some drawbacks, which are worth pointing out.
>   - The first premise of AdamP is that the growth of weights is bad. But this is motivated mainly from the training loss minimization point of view - more specifically, its effect on step sizes. But this does not explain why (generalization) performance deteriorates. As opposed to that view, the argument put forth in our paper is that weight growth leads to premature loss flattening which in turn causes the training process to be stuck in sharper minima with inferior performance. The example in Figure 1 of our paper clearly brings this out.
>   - The second premise of AdamP is that momentum is the main cause of weight growth, and that is motivated mainly via analysis and synthetic simulation of SGD with momentum; see Figure 1 and section 2 of the AdamP paper, which are all about SGD. Unfortunately, the same is not done for Adam and that is because the m_t update is a convex combination of two terms and therefore it does not lead to an extension in length. Of course, weights still grow even for Adam; but that is due to over-parameterization and loss going to very small values. We point out this clearly in our paper. In essence, the second premise of AdamP - that momentum is the cause of weight growth in Adam - is not right.
>   - AdamP says nothing about what it does with the final, fully connected MLP layers of a deep network. At the end of section 3.1 it mentions layer normalization. Also, from the AdamP code (lines 53 and 91 of https://github.com/clovaai/AdamP/blob/master/adamp/adamp.py ) we see that layer normalization is done to all layers, including the final MLP layer(s). This could be a reason why AdamP is sometimes inferior to LAWN, especially at large batch sizes (see results below). By combining free and constrained training and tuning $E_{\text{free}}$ as a hyperparameter, LAWN sets $\{c^\ell\}$, the weight norm values of the layers, to the right values and hence yield great performance. Thus, AdamP raises awareness of weight norm containment and Adam-LAWN effectively tunes it in a data-dependent manner.
>   - One of the downsides of the experimental protocol in the AdamP paper (for example, the comparison of AdamP and AdamW) is that both methods are not tuned independently to yield their best performance. Instead, the methods are only compared at specific values of hyperparameters. In particular, weight decay and learning rate need to be tuned well for each method in order to get the best performance out of a given method. As we will see below, when the hyperparameters are properly tuned for some datasets, AdamW can outperform AdamP.
>
> 2. Our baseline is AdamW and not Adam with L2 regularization
>
> The “Adam” mentioned in all the experiments of section 4 of our paper is actually AdamW (Adam with decoupled weight decay, where we tune the weight decay value). We will describe this clearly when we rewrite the paper.
>
> 3. Tuning $E_{\text{free}}$.
>
> From figure 4, we note that using a small value (in terms of the number of epochs) for $E_{\text{free}}$ works well. We recommend limiting $E_{\text{free}}$ to a value less than 10% of the total epochs and tuning over a grid of three to four values. We note that Adam-LAWN does not add additional hyperparameter tuning complexity when compared to AdamW since we do not use weight decay for Adam-LAWN.
>
> 4. Satisfying the weight norm constraints
>
> In step 11 of Algorithm 1 (Adam-LAWN Constrained phase) we explicitly rescale the weights so that weight norm constraints are satisfied. So eq (3) holds. Again, since this step is hidden inside the algorithm, this is something that we will make more clear where we describe the method in the main text.

---

> ### Author Response · Authors · 2021-11-23
> **Part 2 of response to reviewer Xf6G**
>
> **New experiments comparing Adam-LAWN with AdamP**
>
> We used the PyTorch-based implementation of AdamP (https://github.com/clovaai/AdamP/blob/master/adamp/adamp.py ). We conducted experiments on the CIFAR datasets as well as MovieLens-1M. We tune the learning rate, weight decay and report the average of 3 runs. The AdamW and Adam-LAWN numbers have been taken from the submitted ICLR draft of our paper.
>
> From the results, we conclude the following:
> 1. On CIFAR-10, both AdamP and Adam-LAWN perform similarly.
> 2. On CIFAR-100, AdamP and Adam-LAWN perform similarly for small batch sizes. However, Adam-LAWN is better than AdamP for the large batch setting.
> 3. On MovieLens-1M, Adam-LAWN comprehensively outperforms AdamP across all batch sizes.
>
> It is worth noting that the AdamP paper does not study the effectiveness of AdamP at large batch sizes.
>
> |**Dataset** | **Batch size** | **AdamP** | **AdamW** | **Adam-LAWN** |
> |--|--|--|--|--|
> |CIFAR-10 | 256 | 93.75 | 93.48 | 93.91|
> |CIFAR-10 | 10240 | 93.76 | 92.63 | 93.84|
> |CIFAR-100 | 256 | 72.87 | 70.84 | 72.99 |
> |CIFAR-100 | 10240 | 71.93 | 70.61 | 72.97|
>
> **MovieLens-1M**
>
> |**Batch size** | **AdamP** | **AdamW** | **Adam-LAWN** |
> |--|--|--|--|
> |1k | 69.52 | 69.87 | 70.80 |
> |10k | 68.64 | 70.12 | 70.41 |
> |100k | 68.06 | 69.28 | 70.77 |
> |1M | 66.31 | 68.99 | 70.66 |
>
> **Reference**
>
> 1. Heo B, Chun S, Oh S J, et al. AdamP: Slowing down the slowdown for momentum optimizers on scale-invariant weights[J]. arXiv preprint arXiv:2006.08217, 2020.
> 2. Elad Hoffer, Ron Banner, Itay Golan, and Daniel Soudry. Norm matters: efficient and accurate normalization schemes in deep networks. In Advances in Neural Information Processing Systems, 2018.

---

### Decision · Program_Chairs · 2022-01-20

**Decision:**

Reject

**Comment:**

The reviewers unanimously recommend rejecting this submission and I concur with this recommendation. The submission essentially introduces a regularization technique to solve the alleged problem of Adam getting worse out-of-sample error for typical image classification problems, e.g. training ResNets on ImageNet. Reviewers raised a variety of issues with the submission. Some found the experiments unconvincing, some were concerned that the submission duplicated closely related work without engaging with and citing that work, and some were concerned by what they viewed as insufficient analysis and comparisons. To me, the most severe issue with the submission is that the experimental evidence for its claims is not sufficiently convincing and the problem it purports to solve has not been convincingly demonstrated, making the work hard to motivate. The other issues raised by the reviewers are less damaging in my view.

Although this is a meta review and not a full de novo review, I would be remiss to not raise a few of the severe issues I see with the results that makes it hard for them to be convincing.
The Adam results in table 1 are far weaker than they should be, raising questions about the experiments as a whole. For example, https://arxiv.org/abs/2102.06356 reports 76.4% top 1 accuracy for ResNet-50 on ImageNet with Adam without increasing the epsilon parameter to a larger value as Choi et al. 2019 did (who also report good Adam results for ResNet-50 on ImageNet). This should also lead us to question one premise of the paper that there is some problem with adaptive optimizers for image classification.

Ok, but perhaps LAWN helps validation error even if there is no gap between SGD and Adam? Sadly, to demonstrate this subordinate claim, LAWN would have to be compared carefully with state of the art regularization techniques and compared with results that use any optimizer, not just Adam. With modern regularization techniques, it isn't hard to get 77%+ top 1 validation accuracy on ImageNet with ResNet-50. See, for example https://arxiv.org/abs/2010.01412v1 which gets 77.5% in 100 epochs and as high as 79.1 with longer training. Since LAWN is claiming to improve generalization, it must be compared with other regularization techniques. It is a type error to primarily compare it with optimizers so even if there weren't concerns with the performance of the existing baselines, there would need to be additional comparisons.

The claims about fixing issues that arise at large batch sizes are prima facie problematic since there isn't strong evidence of an actual problem at the batch sizes considered in the submission.